# Histological characterization of anther structure in Tetep-cytoplasmic male sterility and fine mapping of *restorer-of-fertility* gene in rice

Seung Young Lee[1], Zhuo Jin[2], Su Jang[1], Backki Kim[1], Jeonghwan Seo[3], Hee-Jong Koh[1] *

1 Department of Agriculture, Forestry and Bioresources, Plant Genomics and Breeding Institute, Research Institute for Agriculture and Life Sciences, Seoul National University, Seoul, Korea, 2 Department of Integrative Biological Sciences and Industry, Sejong University, Seoul, Korea, 3 National Institute of Crop Science, Rural Development Administration, Wanju, Korea

* heejkoh@snu.ac.kr

**Data Availability Statement:** The Next Generation Sequencing (NGS) raw data described in this study are openly available in the NCBI Short Read Archive (SRA) databases under following BioProject

## Abstract

Cytoplasmic male sterility (CMS) is a maternally inherited trait that inhibits plants from producing or releasing viable pollen. CMS is caused by mitochondrial–nuclear interaction, and can be rescued by introducing functional nuclear *restorer-of-fertility* (*Rf*) gene. The Tetep-CMS/*Rf* lines were developed through successive inter-subspecific backcrosses between *indica* and *japonica* rice accessions. Phenotypic characterization of Tetep-CMS lines revealed abnormal anther dehiscence and the inability to release, while possessing functional pollen. Transverse sections of developing anthers collected from CMS plants showed connective tissue deformities and aberrant dehydration of endothecium and epidermis. Fine mapping of *Rf*-Tetep using a series of segregating populations, delimited the candidate region to an approximately 109 kb genomic interval between M2099 and FM07 flanking markers. Nanopore long-read sequencing and genome assembly, proceeded by gene prediction and annotation revealed 11 open reading frames (ORFs) within the candidate region, and suggest ORF6 annotated as pentatricopeptide repeat motif containing gene 1 (*PPR1*), as a possible candidate gene responsible for fertility restoration. This study suggests that tissue-specific abnormalities in anthers are responsible for indehiscence-based sterility, and propose that the functional *Rf* gene is derived from allelic variation between inter-subspecies in rice.

## Introduction

Cytoplasmic male sterility (CMS) is a maternally inherited trait, phenotypically expressed as the inability to produce functional pollen [1]. CMS has been observed in over 150 plant species derived either from inter-subspecific crosses or by mutagenesis [2]. The CMS-inducing genes are often associated with chimeric mitochondrial open reading frames (ORFs), and their

accession numbers: HopumR nanopore (PRJNA797641), HopumR Illumina (PRJNA797356), Tetep nanopore (PRJNA797573), Hopum nanopore (PRJNA797574).

**Funding:** This study was supported by a grant from the Next-Generation BioGreen 21 Program (No. PJ013165) of the Rural Development Administration, Korea.

**Competing interests:** The authors have declared that no competing interests exist.

expression can be suppressed when combined with the nuclear *restorer-of-fertility* (*Rf*) genes [3]. Therefore, the CMS/*Rf* system is an ideal genetic resource to study interaction between mitochondrial gene and nuclear genome for hybrid crop production.

The mode of action of CMS-inducing mitochondrial genes is quite complex, and several studies have been conducted to elucidate the effect of mitochondrial genome rearrangements on CMS. Previous reports showed that CMS systems exhibit spatiotemporal accumulation of CMS proteins, potentially resulting in cytotoxicity, energy deficiency and premature programmed cell death (PCD), leading to alterations in either gametophytic or sporophytic cells in male organs [4, 5]. To date, nine types of CMS systems have been discovered in rice: *orf79* from BT-CMS (Chinsurah Boro II) [6]; *orf307* from CW-CMS (Chinese wild) [7]; *orf182* from D1-CMS (Dongxiang) [8]; *orf290* and *orfH79* from HL-CMS (Honglian) [9, 10]; *L-orf79* from LD-CMS (Lead rice) [11]; *orf113* from RT98-CMS (RT98C) [12]; *orf352* from RT102-CMS (RT102C) [13]; *orf312* from TA-CMS (Tadukan) and Tetep-CMS [14, 15] and *WA352c* from WA-CMS (Wild abortive) [16]. Correlations among the anther developmental stage, pollen morphology, and abortion time signify the feasible target where the localization of chimeric CMS genes occur. In WA-CMS, WA352c encodes a transmembrane protein which accumulates in anthers at the microspore mother cell stage and degenerates tapetum cells resulting in the production of aborted pollen grains at the early uninucleate stage [16, 17]. By contrast, pollen abortion in BT-CMS and LD-CMS system occurs at the trinucleate stage. Therefore, the pollen stain slightly with $I_2$-KI and are spherical in shape, but exhibit starch deficient characteristics, which makes them non-functional [11].

Restorer lines possess functional nuclear *Rf* genes, which encode mitochondria-targeted proteins that suppress the CMS-related defects at the transcriptional, translational and retrograde regulatory levels [4]. Different *Rf* genes have been identified to restore particular CMS systems in rice: *Rf1a* and *Rf1b* for BT-CMS [6]; *Rf17* for CW-CMS [18]; *Rf5* and *Rf6* for HL-CMS [19]; *Rf2* for LD-CMS [20]; *Rf98* for RT98-CMS [21]; *Rf102* for RT102-CMS [13]; *Rf3* and *Rf4* for WA-CMS [22, 23]. Based on the properties of encoded proteins, more than half of the known *Rf* genes, including *Rf1a* (*Rf5*), *Rf1b*, *Rf4*, *Rf6* and *Rf98*, have been annotated as pentatricopeptide repeat (PPR) motif-containing proteins. PPR proteins harbor 35-amino acid residues present as multiple (2–30) tandem repeats [24]. These specific RNA-binding proteins are known to restore fertility by decreasing the transcript levels of CMS-related genes or by suppressing their expression via gene editing, splicing or by cleaving the dicistronic transcript between the CMS gene and the interaction domain [6, 25–27].

Previously, we developed the Tetep-CMS system through successive rounds of backcrosses between rice cultivars belonging to two different subspecies, Tetep (*indica* rice) and Hopum (*japonica* rice) cultivars. Unlike other CMS types, which exhibit non-functional pollen phenotypes, Tetep-CMS exhibits abnormal anther dehiscence while possessing functional pollen. Whole genome sequencing and de novo assembly of the mitochondrial genome of Tetep-CMS lines revealed the presence of a chimeric gene, *orf312*, with COX11 as the interaction domain. We further identified a quantitative trait locus (QTL) for *Rf*-Tetep on the long arm of chromosome 10, which harbors a cluster of several *Rf* and *Rf*-like (RFL) genes [15].

In this study, we characterized the anther dehiscence phenotypes of the Tetep-CMS line, and performed transverse sectioning of anthers to identify the cause of abnormal anther dehiscence. We performed nanopore long-read sequencing-based genome assembly to obtain genomic information and conducted fine mapping using a series of segregating populations. This led to the identification of a genomic region, which co-segregated with the fertility phenotype, thus suggesting a candidate gene that could potentially restore fertility in the Tetep-CMS line. This result may help to resolve the genetic vulnerability of CMS/*Rf* systems and enhance our understanding of the mitochondrial–nuclear genome interaction in hybrid rice.

## Materials and methods

### Plant materials

Rice accession Hopum A (Tetep-CMS; *rfrf*) was crossed with a $BC_3F_2$ line (*Rfrf*; recurrent parent) derived from an inter-subspecific cross between Tetep (*indica* rice; *RfRf*) and Hopum (*japonica* rice; *rfrf*) cultivars. Through successive selfing along with genotypic and phenotypic screening (n = 2,217), a total of 576 $F_3$ recombinant individuals were used for fine mapping to delimit the *Rf*-Tetep candidate region.

Hopum R (restorer line), Tetep (*Rf* donor), and Hopum (maintainer line) were subjected to Oxford nanopore technology (ONT) long-read sequencing to assess genomic structural variation and sequence the candidate region. All plant materials were cultivated in a paddy field and artificial crosses were conducted in a greenhouse at the experimental farm of Seoul National University, Suwon, Korea. Detailed pedigree of all plant materials are illustrated in S1 Fig.

Agronomic traits and panicle characteristics were measured 115 days after transplanting (DAT). Mature anthers of Tetep-CMS/*Rf* lines were harvested 24 h after anthesis and pollen release was observed after the complete dehydration and staining of anthers using absolute ethanol and 0.1% (*w/v*) potassium iodide ($I_2$-KI), respectively. All anther samples were observed using CX31 light microscope (Olympus, Japan), and photographed using eXcope T500 microscope camera (DIXI Science, Korea).

### Histological assay

Spikelets were sampled at different stages of anthesis. Immediately after removing the lemmas, spikelets were fixed in 4% paraformaldehyde (PFA) at 4°C for 24 h. The fixed spikelets were dehydrated by immersing in a series of ethanol concentrations (30%, 50%, 70%, 85%, 95% and 100%) for 30 min at each concentration. After complete dehydration, the samples were soaked in ethanol: Histo-Clear II (National diagnostics, USA) (3:1, 1:1 and 1:3 ratios [*v/v*]) for 30 min, and then in 100% Histo-Clear II for 24 h. To infiltrate the samples with paraffin, Paraplast Plus (Sigma, USA) was gradually added with Histo-Clear II solution (3:1, 1:1 and 1:3). After a series of infiltrations, the samples were stored in 100% paraffin at 58°C for 24 h. The paraffin-infiltrated samples were embedded in an embed block, and sectioned into 10 µm with Microm HM 325 Rotary Microtome (Thermo Scientific, USA). Subsequently, the sections were stained with 1% toluidine blue (Sigma, USA) and observed under CX31 light microscope (Olympus, Japan). Images were captured using eXcope T500 microscope camera (DIXI Science, Korea).

### Marker development and fine mapping

Genomic DNA was extracted from the young leaves using the modified cetyltrimethylammonium bromide (CTAB) method [28]. The concentration and purity of each DNA sample were measured with NanoDrop 1000 spectrophotometer (NanoDrop Technologies, USA). To identify nucleotide sequence polymorphisms, 17 molecular markers (S1 Table) were designed using Primer3 version 0.4.0 [29], based on pseudomolecules assembled with long-read sequencing data. Whole genome sequencing data of two parental cultivars, Hopum (PRJNA705813) and Tetep (PRJNA705829), retrieved from the Sequence Read Archive (SRA) database of the National Center for Biotechnology Information (NCBI), were used to validate the marker [15]. PCR was performed in a 20 µl volume containing approximately 100 ng of gDNA template, 2 µl of 10X PCR buffer, 1 µl of dNTPs (10 mM), 1 µl of each primer (10 µM) and 0.5 U of Prime *Taq* polymerase (GeNet Bio, Korea). The thermocycling conditions were

as follows: initial denaturation at 95 ˚C for 10 min, 35 cycles of 95˚C for 30 s, annealing at 57˚C for 30 s, 72˚C for 30 s and final extension at 72˚C for 10 min.

## ONT long-read and whole genome sequencing

High molecular weight genomic (HMW) DNA was extracted from the young leaves of Hopum R, Tetep, and Hopum. Pre-cooled mortar and pestle was used to grind fresh young leaves in liquid nitrogen and transferred to Carlson lysis buffer (100 mM Tris-HCl, pH [9.5], 2% CTAB, 1.4 M NaCl, 1% PEG 8000 and 20mM EDTA), according to the HMW gDNA extraction protocol (Oxford Nanopore Technologies, UK). The isolated HMW gDNA was purified using the genomic tip 100/G (Qiagen, Germany) according to the manufacturer's instructions. Nanopore long-read sequencing was performed using GridION platform at Phyzen (Phyzen Inc., Korea). Adapter sequences at the end of the ONT sequencing reads were trimmed using Porechop [30].

To perform whole genome sequencing, DNA libraries were constructed using the TruSeq Nano DNA kit (Illumina, USA) from gDNA samples prepared for ONT long-read sequencing. The DNA libraries were sequenced on the Illumina NovaSeq 6000 platform in paired-end mode at Macrogen (Macrogen Korea, Korea).

## Genome assembly, polishing and scaffolding

Genomes of Hopum R, Tetep, and Hopum were assembled according to the modified workflow of nanopore sequencing previously used to assemble the genome of circum-basmati rice [31]. Raw nanopore sequence reads were corrected and assembled using NextDenovo (https://github.com/Nextomics/NextDenovo). Raw sequence reads shorter than 8 kb in length were filtered out using the parameter "read_cutoff = 8k". The draft assemblies were polished for two rounds with NextPolish [32] using the whole genome sequencing data. The quality of genome assembly was assessed using QUAST-5.0.2 [33]. The completeness of the genome assembly was calculated using BUSCO v5.0.0, with the lineage dataset of embryophyta_odb10 [34]. Nipponbare as the reference genome [35], we scaffolded the contigs using reference genome-guided scaffolding tool, RagTag [36]. Synteny between the assembled genome sequences and the Nipponbare reference genome was visualized as dot plots using D-Genies [37].

## Gene annotation and structural variation assessment

Nucleotide sequence of the candidate region was extracted using the bedtools getfasta option [38]. Genes were predicted with Augustus ver. 2.5.5 using rice gene model as the training set [39]. Gene structures were validated using NCBI ORFfinder [40] and annotated using local blastn [41] search against the IRGSP-1.0 gene sequences in the Rice Annotation Project Database (RAP-DB) [42]. PPR domains were determined using NCBI Conserved Domain Search [43] and ScanProsite [44]. Sequences of chromosome 10 of Hopum R, Tetep, and Hopum were extracted from the corresponding genome assemblies, and structural variations were examined using the multiple genome alignment tool MAUVE [45].

## Allelic variation and distribution of *Rf*-Tetep

The allelic variation between the *Rf* alleles of Tetep and Hopum were identified by aligning the genome sequences of the two cultivars using ClustalW (http://www.genome.jp/tools-bin/clustalw). Single nucleotide polymorphisms (SNPs) were retrieved and haplotype network analysis was conducted using RiceVarMap [46]. To detect the presence of *orf312* in accessions sharing identical genotypes as Hopum R and Tetep, the raw whole genome sequencing data

were obtained from the 3K Rice Genome Project [47] and quality controlled using fastp [48]. Sequence reads originating from the mitochondrial genome were extracted using BWA-MEM2 and SAMtools [49, 50]. The mitochondrial reads were then subjected to de novo assembly using SPAdes [51]. The *orf312* gene was searched using local blastn [41].

## Results

### Phenotypic characterization of Tetep-CMS and restorer lines

The Hopum A, Hopum R, and Hopum are near-isogenic lines (NILs) with no significant differences with $F_1$ on their plant height, panicle number, and panicle length (Fig 1A and S2 Table). Hopum A was completely sterile, whereas the other lines were fertile (Fig 1B). The panicle sterility of Hopum A was mainly caused by abnormal anther dehiscence, whereas $F_1$ lines, Hopum R, and Hopum exhibited fully dehisced anthers in the apical and basal portion of the anther (Fig 1C). The release of pollen was observed by $I_2$-KI staining of anthers collected 24 h after flowering. The $F_1$ lines, Hopum R, and Hopum showed normal anther dehiscence, as indicated by proper pollen release; on the other hand, Hopum A pollen were contained within the anther, and inability of the pollen release was observed (Fig 1D).

### Histological analysis of anthers at different developmental stages

The hypothetical association between the abnormal dehiscence phenotype and anatomical structure of anthers was tested by examining the transverse sections of anthers sampled at various developmental stages. Significant differences in the connective tissue were observed between fertile and sterile lines during early anthesis (Fig 2A and 2B). Because of cell differentiation, connective tissue of Hopum R showed an organized pattern of enlarged cells, whereas Hopum A showed relatively small cells arranged in a dense and unorganized pattern. At the mid-anthesis stage, two developmental changes were observed, that are known to be essential for proper anther dehiscence. Firstly, a cavity for dehiscence was formed, which was accompanied by the appearance of stomium (breakage site for anther dehiscence) and septum (cell layer between the connective tissue and locule). Secondly, because of dehydration of the outer layer of anthers, the epidermis and endothecium exhibited a shrunken phenotype (Fig 2C) [52]. However, Hopum A anthers seemed to be developmentally arrested at the first step (cavity formation for dehiscence), and did not show any signs of dehydration in the outer layer (Fig 2D). During progression from late anthesis to the anthesis stage, Hopum R anthers showed the complete breakage of septum and expansion of locule, in accordance with the rapid emergence of stomium (Fig 2E). At the anthesis stage, Hopum R anthers showed complete rupture of stomium, which was simultaneously accompanied by pollen release (Fig 2G). By contrast, Hopum A anthers showed no complementary developmental events as Hopum R anthers at the late anthesis and anthesis stages; instead, Hopum A anthers were developmentally arrested at the cavity formation step and showed only slight shrinkage of the outer layer (Fig 2F and 2H). To determine the relationship between abnormal dehiscence and water transport, the filament that connects the anther to the plant body was further sectioned. No remarkable structural differences were observed between Hopum R and Hopum A (S2 Fig), indicating that the structural abnormalities occur independently within the Hopum A anthers and not affected by external factors such as water and nutrient transport.

### Fine mapping of *Rf*-Tetep

To map the *Rf*-Tetep locus, we conducted bulked segregant analysis (BSA) and QTL-seq of the segregating $BC_3F_1$ population, and identified a QTL on the long arm of chromosome 10 at a

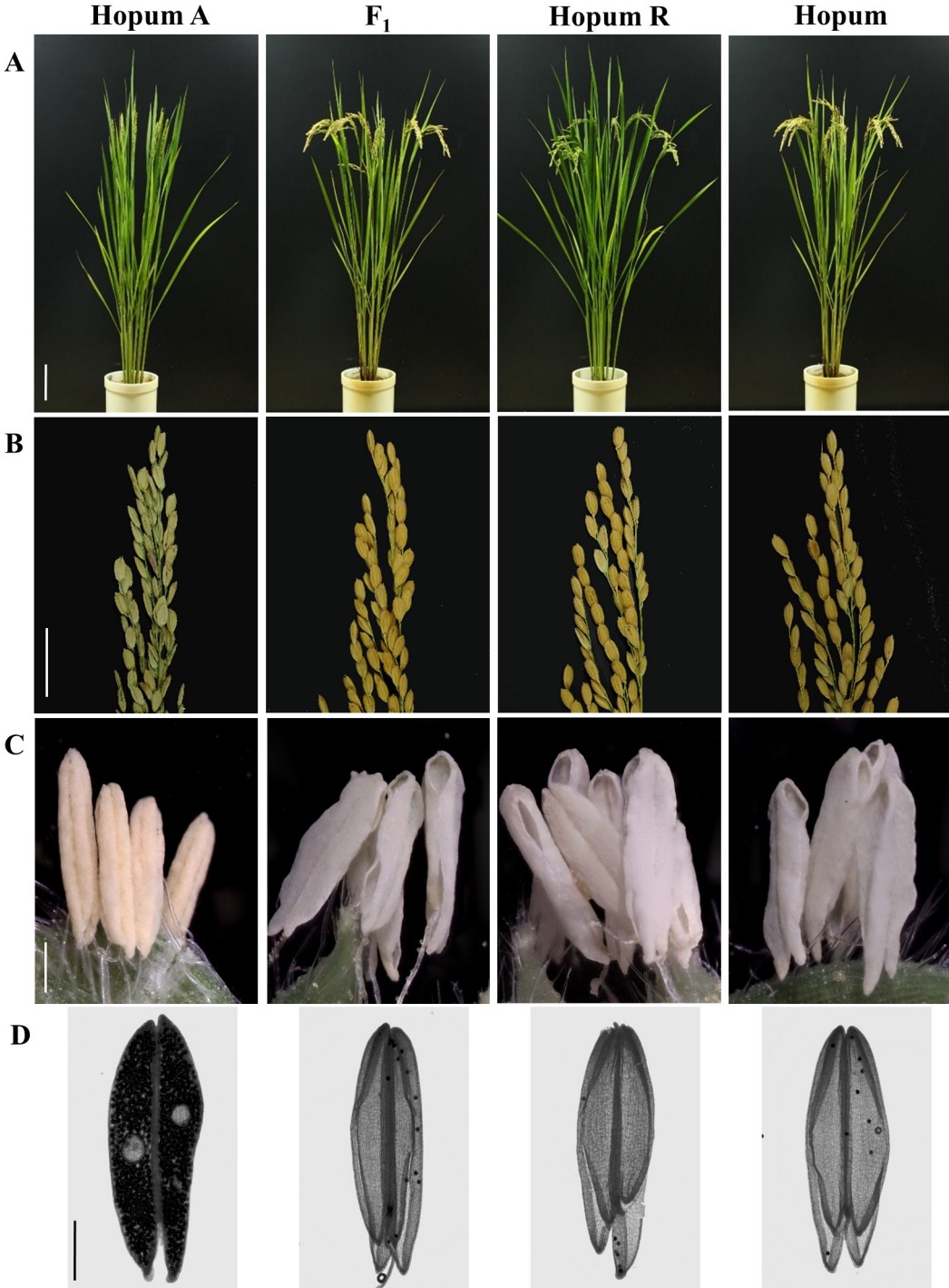

**Fig 1. Phenotypic comparisons among the Tetep-CMS line, restorer line, and their F$_1$ progeny.** (A) Plant phenotype before harvest. Scale bar = 10 cm. (B) Seed setting. Scale bar = 2 cm. (C) Anthers 24 h after flowering. Scale bar = 500 μm. (D) 0.1% I$_2$-KI stain after flowering. Scale bar = 500 μm.

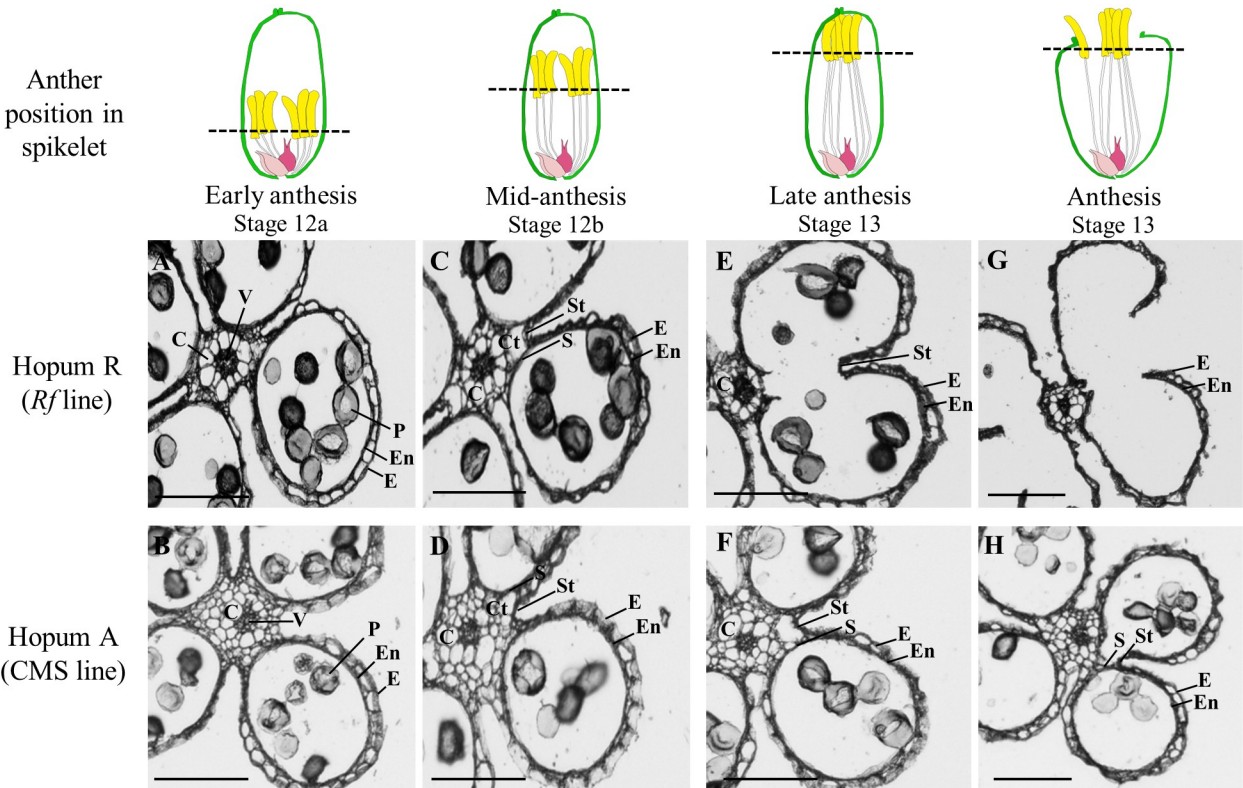

**Fig 2. Transverse sections of Hopum R and Hopum A anthers at different stages of anthesis.** (A, B) Early anthesis: Stage 12a. (C, D) Mid-anthesis: Stage 12b. (E, F) Late anthesis: Stage 13. (G, H) Anthesis: Stage 14. C: connective tissue; V: vascular bundle; M: mature pollen; En: endothecium; E: epidermis; Ct: cavity for dehiscence; S: septum; St: stomium. Dotted lines indicate the position of sectioning. Scale bars = 250 μm.

physical distance of 16–24 Mb [15]. To further narrow down the candidate region of *Rf*-Tetep, we conducted primary fine mapping using a total of 2,217 $F_1$ and $F_2$ individuals (Fig 3A), and selected 576 $F_3$ recombinants for the final fine mapping (Fig 3B). The co-segregation of plant genotype and phenotype resulted in the identification of a genomic interval between flanking markers M2099 and FM07 (Fig 3C). This genomic interval was located between 18.68 and 18.79 Mb (IRGSP 1.0), which narrowed down the candidate region to approximately 109 kb. Individuals possessing Hopum allele within the candidate region exhibited seed setting rate of 0–0.35%, whereas those harboring the Tetep allele showed a seed setting rate of 52.26–85.86% (Fig 3D).

## ONT long-read sequencing and genome assembly

Using the ONT GridION platform, the long-read sequencing of genomes generated 3,978,209 reads (19.28 Gb) for Hopum R, 3,350,388 reads (14.58 Gb) for Tetep, and 3, 277,590 reads (14.55 Gb) for Hopum. The average read lengths of Hopum R, Tetep, and Hopum were 4,847, 4,351, and 4,439 bp, respectively, and the N50 length was >8.6 kb for all samples (S3 Table). The implementation of read cutoff at 8 kb resulted in the selection of 693,021 reads of Hopum R, 494,942 reads of Tetep, and 504,546 reads of Hopum, which was equivalent to a genome coverage depth of 36.30×, 26.41×, and 25.68×, respectively. We polished the genome assemblies with short Illumina paired-end sequence reads. The polished genome assemblies spanned 374.3 Mb across 176 contigs for Hopum R, 383.9 Mb across 232 contigs for Tetep, and 371.6

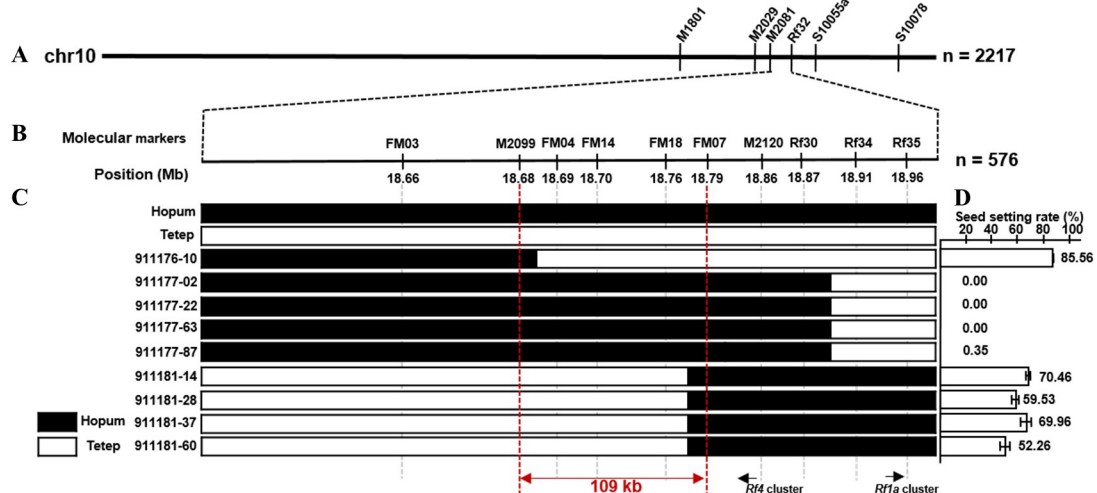

**Fig 3. Fine mapping of *Rf*-Tetep using F$_1$, F$_2$ and F$_3$ populations.** (A) Genetic mapping using F$_1$ and F$_2$ populations. (B) Molecular markers and its corresponding genomic positions (IRGSP-1.0). (C) Delimitation of the candidate region using F$_3$ individuals. (D) Seed setting rate (%). Black and white blocks indicate Hopum and Tetep alleles, respectively.

Mb across 237 contigs for Hopum. Our draft genome assemblies were aligned against the Nipponbare (*japonica* rice) reference genome using the reference guided scaffolding tool RagTag. Our final genome assemblies recovered >98% of 1,440 BUSCO embryophyte lineage gene groups; this BUSCO score was of good quality as it exceeded 95% recovery (S4 Table).

To assess the syntenic relationship among CMS lines, we aligned the assembled genome sequences of Hopum R, Tetep, and Hopum against the Nipponbare reference genome [35]. The dot plot showed collinearity between the assembled genomes and the reference genome over all chromosomes (Fig 4A). Further genome sequence analysis at the chromosome level revealed several large gaps on chromosome 10 of Hopum R (8.29–9.36, 11.05–11.47, and 12.83–13.61 Mb), Tetep (8.12–10.61 and 12.07–12.49 Mb), and Hopum (1.44–2.51 and 4.38–8.09 Mb). However, our region of interest (18.50–18.98 Mb) showed macrosynteny and high level of sequence identity (Fig 4B). Since the main objective of constructing these genome assemblies was to obtain the genetic information of each line rather than completing the genome using de novo method, we decided to annotate the genes located within the region of interest and then compare the gene sequences to identify structural variations.

## Multiple sequence alignment and candidate gene analysis

We initially assumed that genomic segments that are both present in Tetep and Hopum R and absent in Nipponbare and Hopum genomes have a great chance of possessing *Rf*, an *indica*-specific gene derived from Tetep cultivar. Multiple sequence alignment using MAUVE revealed two remarkable insertions in chromosome 10 of Hopum R (19.83–19.88 and 20.01–20.25 Mb). Gene prediction, annotation and conserved domain search revealed six *PPR* genes within the above mentioned regions: *PPR780*, *PPR796*, *PPR931*, *PPR683*, *PPR762*, and *PPR777* (Fig 5A). According to evolutionary plasticity studies, *PPR683* and *PPR762* are *Rf*-like genes common among *indica* accessions [53]; *PPR780* was previously annotated as *PPR8-780-M* as *Rf4*-like gene [27]; and *PPR777* was reported to be derived from the W1109 strain of wild rice *Oryza rufipogon*, the ancestor of present-day cultivated rice [21]. Notably, *PPR796* and *PPR931* are as yet unreported *PPRs*, discovered for the first time in this study.

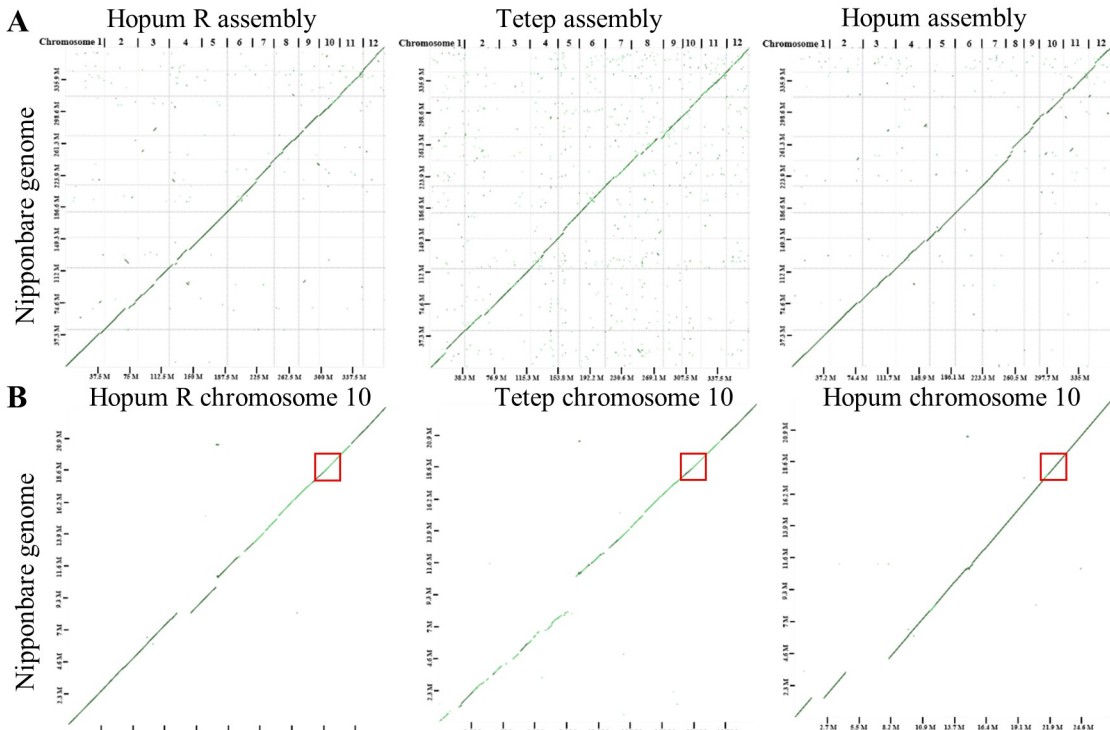

**Fig 4. D-GENIES dot plots showing the syntenic relationships between Nipponbare reference genome and the assembled genomes.** (A) All chromosome of Hopum R, Tetep, and Hopum. (B) Only chromosome 10. Red box indicates the genomic region of interest.

However, when fine mapping results were applied to our genome sequence alignments, neither of the two regions were included in the candidate region. Therefore, we provisionally concluded that the *Rf* gene was not derived from Tetep-specific genome segments, and functions as a *Rf* gene because of allelic differences between Hopum and Tetep. According to the Michigan State University (MSU) Rice Genome Annotation Project [42] and RAP-DB [54], 11 functional genes were predicted within the 109 kb candidate region (Fig 5B) (Table 1). Among

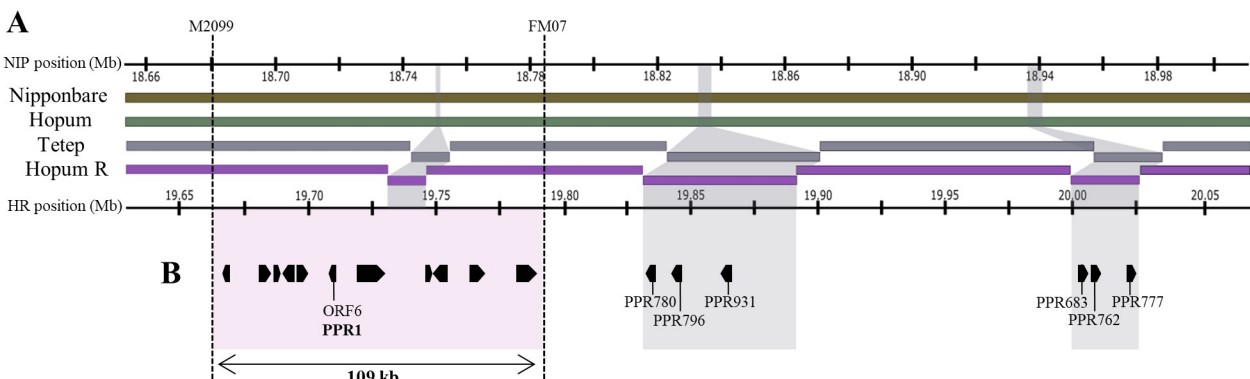

**Fig 5. Synteny and gene annotation within the candidate region.** (A) Multiple sequence alignment of Nipponbare, Hopum, Tetep and Hopum R using MAUVE. (B) Illustration of 11 candidate genes annotated within the 109 kb candidate region (red portion), and PPR genes annotated from Tetep-specific insertions (grey portion).

**Table 1. Candidate genes in the genomic region between M2099 and FM07 markers.**

| ORF | Gene ID | Description |
|---|---|---|
| ORF1 | Os10g0492300 | Similar to IAP100 |
| ORF2 | Os10g0492400 | Similar to protein kinase domain containing protein |
| ORF3 | Os10g0492600 | Similar to tonoplast membrane integral protein ZmTIP3-1; probable aquaporin TIP3.1 |
| ORF4 | Os10g0492800 | Similar to Ser/Thr protein phosphatase family |
| ORF5 | Os10g0492900 | Similar to alpha-galactosidase |
| ORF6 | Os10g0493100 | Pentatricopeptide repeat domain containing protein, PPR1, K homology; type 1 domain containing protein; similar to KH domain containing protein |
| ORF7 | Os10g0493600 | Alpha-galactosidase precursor (EC 3.2.1.22) (Melibiase) (Alpha-D- galactoside galactohydrolase) |
| ORF8 | Os10g0493900 | Protein of unknown function DUF6; transmembrane domain containing protein |
| ORF9 | Os10g0494000 | Protein of unknown function DUF789 family protein |
| ORF10 | Os10g0494200 | Similar to N-acetyl-gamma-glutamyl-phosphate reductase |
| ORF11 | Os10g0494300 | ATP binding cassette G transporter; regulation of male reproduction; anther cuticle development |

those 11 ORFs, ORF7 and ORF8 were identical among Hopum R, Tetep, and Hopum, while the remaining ORFs showed Tetep-specific alleles, which were identical to Hopum R alleles and different from Hopum alleles. We also observed an unannotated Tetep-specific segment with features of a retrotransposon, within the region. *Rf* genes with PPR motifs are expected to encode mitochondrial proteins. By rapidly screening the proteomes for N-terminal sequences using Predotar v1.04 [55], we identified candidate genes that possibly targets mitochondria. Therefore, we suggest ORF6, annotated as *PPR1*, as a probable candidate gene for *Rf*-Tetep.

## *PPR1* allelic variation and geographic distribution

To identify the natural variation of *PPR1*, we first extracted the genomic region of *PPR1* from Tetep and Hopum. Two SNPs were detected in the PPR1 coding sequence between the two cultivars, corresponding to two non-synonymous substitutions: glycine (G) to aspartic acid (D) and histidine (H) to glutamine (Q). In addition, two SNPs were detected in the in 3' untranslated region (3'UTR) and one in 5' untranslated region (5'UTR) of *PPR1* (Fig 6A). Pairwise alignment of Hopum and Tetep *PPR1* amino acid sequences revealed that the G→D substitution was located within the PPR motif 1 (S3 Fig). To trace the distribution pattern of *PPR1*, we input the SNP variation IDs into the haplotype network analysis of RiceVarMap2 [46], which contains information on 41,709 SNPs among 4,729 rice accessions. The result of haplotype network analysis showed that *PPR1* diverged into four types and revealed that Tetep allele belongs to the most abundant type 1 group (Fig 6B). When we screened for accessions possessing both type 1 genotype and *orf312* sequences, a total of 134 accessions were detected, which diverged within the *indica* sub-classes: 7 accessions in *indica* I, 3 in *indica* II, 78 in *indica* III, and 46 in *indica* admixture (Fig 6C and S5 Table). On the other hand, *orf312* was not detected in the accessions possessing the type 2 genotype. Accessions possessing both type 1 genotype and *orf312* were mostly concentrated in Asia, while few were found in the Africa and American continents (Fig 6D). These accessions could be used as genetic resources in the future to develop restorer lines for the Tetep-CMS system.

## Discussion

Anther dehiscence is an essential process for plant reproduction as it allows the release of functional pollen at the flowering stage, thus enabling the pollination and fertilization [56]. Unlike

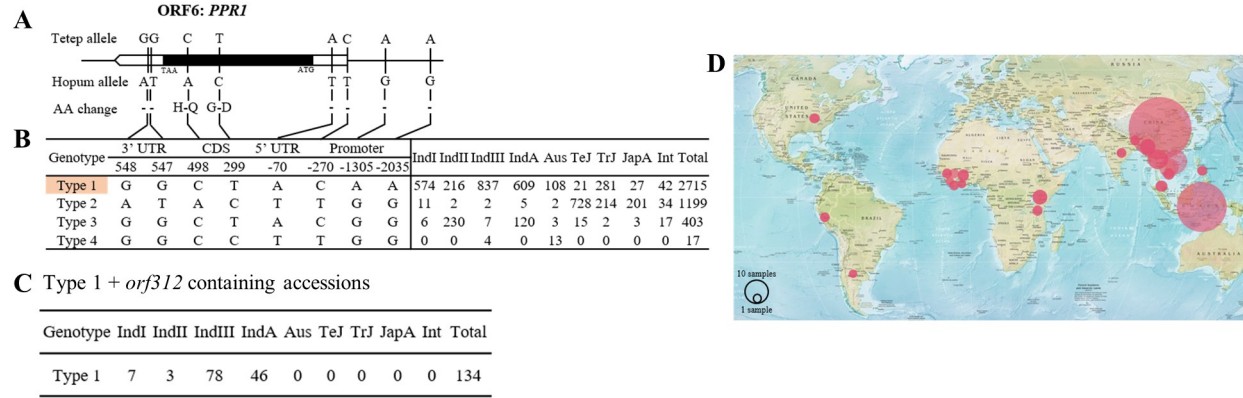

**Fig 6. Haplotype network analysis of *PPR1*.** (A) Schematic of the *PPR1* gene structure and allelic variation between Hopum and Tetep. (B) Haplotype analysis of the PPR1 gene region, based on 4,729 rice cultivars. (C) Number of accessions possessing both type 1 genotype and *orf312*. (D) Geographical distribution of *PPR1*-Tetep allele-containing accessions. The map was retrieved from Central Intelligence Agency (CIA) Factbook for illustrative purposes only.

most CMS systems, where sterility is caused by non-functional pollen, the Tetep-CMS line possesses viable pollen but lacks the ability to rupture anthers, leading to panicle sterility. In the present study, we discovered that sterility in the Tetep-CMS system was caused by abnormal anatomical characteristics of the connective tissue and outer layer, along with the malfunction of stomium and septum during anthesis. The *orf312*-encoded peptide is toxic to *Escherichia coli*, which potentially indicates the possibility of accumulation of reactive oxygen species (ROS) [15, 57]. ROS function as a critical signaling molecules during developmental PCD, leading to post-translational and transcriptional modifications [58, 59]. In WA-CMS, inhibition of ROS-scavenging is observed when WA352c targets COX11 and suppresses its expression, leading to premature tapetal PCD, which results in pollen abortion because of the lack of nutrient supply to microspores [16]. Similarly, disturbed timing of PCD in other sporophytic tissues such as connective tissue, endothecium, and epidermis, may enable proper tapetal degeneration and microspore development, consequently resulting in the inefficient release of viable pollen. [60–63]. Dehydration leads to the shrinkage of endothecium and epidermis, which allows the locule to push outward, resulting in the complete rupture of septum and stomium. However, the anther dehiscence program of Hopum A is possibly hindered by premature PCD in connective tissues, where the absence of cell differentiation and degeneration result in the maintenance of the anther structure in the early developmental state and enable the continuous supply of water along the outer layer of the anther [64]. This dense and unorganized arrangement of cells in connective tissue alters the endothecium and epidermis to preserve water status and prevents them from driving the outward pressure of the locule, thus avoiding the final split of the stomium.

Identification of *Rf*-Tetep which possibly rescues the *orf312*-COX11 transcript, was conducted via fine mapping using segregating populations. We delimited the candidate region to approximately 109 kb on the long arm of chromosome 10 between M2099 and FM07 markers. We then aligned the genomic sequences of the candidate region in Hopum R, Hopum, Tetep, and Nipponbare, and predicted 11 ORFs. Among those ORFs, we proposed ORF6, annotated as *PPR1* as the possible candidate gene. PPR motif-containing proteins bind to mRNAs in a sequence specific manner, thus regulating gene expression [24]. Previous studies on *Rf* genes in rice demonstrated that PPR proteins suppress the detrimental effects of CMS-related proteins by modifying the transcription of corresponding genes. The expression of CMS-related

defects is reduced when *Rf4* suppresses the expression of WA352c and *Rf1b* post-transcriptionally degrades the atp6-*orf79* transcript [5, 6, 27]. Similarly, two PPR motifs in *PPR1* could bind to the *orf312* transcript and repress its expression. However, whether PPR1 forms a functional complex with additional co-factors, because of absence of endonuclease activity, needs to be further elucidated [27, 65].

Comparison of *PPR1* between Tetep and Hopum revealed two SNPs within its coding region, both of which led to non-synonymous amino acid substitutions, and one of these substitutions (glycine to aspartic acid) was located within the PPR motif. The result of haplotype network analysis indicated that type 1 allele was the most abundant among *indica* accessions. In general, hybrid rice accessions derived from intra-subspecific crosses exhibit higher general combining ability (GCA) than those derived from inter-subspecific crosses [66, 67]. The incompatibility between *indica* and *japonica* hybrids is affected by a large number of loci, namely *S5* and *S24*, responsible for hybrid sterility [68]. Therefore, rice genotypes must be pre-screened by marker-assisted selection (MAS) to identify *indica* accessions that are compatible with *japonica* accessions. Thus, the high frequency of type 1 allele is not equitable enough to signify the number of functional *Rf* alleles. To assess their genetic similarity with the Tetep cultivar, we first screened for accessions carrying both type 1 allele and *orf312*. The subpopulation of screened accessions was limited to the *indica* type and was mainly distributed in tropic and sub-tropic regions. These accessions have advantages of scoring good combining ability with *japonica* cultivars and could be used to develop *indica*-compatible *japonica* restorer lines.

## Conclusion

In this study, we phenotypically characterized the Tetep-CMS line and discovered the relationship between the abnormal dehiscence phenotype and anatomical structure of anthers during development. We also suggested a candidate *Rf* gene, which functions potentially because of allelic differences between two parental rice subspecies. These results will serve as a stepping stone for determining the variation of sporophytic PCD in CMS and provide options for the development of CMS/*Rf* systems.

## Supporting information

**S1 Fig. Pedigree of plant materials used in this study.** Solid arrow indicates the progeny of a single cross. Dotted arrow indicates successive rounds of crosses. The X mark enclosed within a circle indicates selfing.
(TIF)

**S2 Fig. Transverse section of filament.** Scale bars = 100 μm. Dotted line represents the position of sectioning.
(TIF)

**S3 Fig. Pairwise alignment of PPR1 amino acid sequences of Tetep and Hopum.** Shaded portion represents the two PPR motifs.
(TIF)

**S1 Table. List of molecular markers used for fine mapping.**
(DOCX)

**S2 Table. Agronomic traits of the Tetep-CMS line.**
(DOCX)

**S3 Table. Raw data obtained by nanopore long-read sequencing.**
(DOCX)

**S4 Table. Summary of genome assemblies.**
(DOCX)

**S5 Table. List of accessions possessing both type 1 genotype and *orf312* sequences.**
(DOCX)

# Acknowledgments

We thank the members of Crop Molecular Breeding Laboratory at Seoul National University for their assistance and guidance. We also express our sincere gratitude to Dr. Hong-Yeol Kim for managing the experimental field.

# Author Contributions

**Conceptualization:** Hee-Jong Koh.

**Data curation:** Seung Young Lee, Zhuo Jin, Su Jang, Backki Kim, Jeonghwan Seo.

**Formal analysis:** Seung Young Lee.

**Funding acquisition:** Hee-Jong Koh.

**Investigation:** Seung Young Lee, Zhuo Jin, Su Jang, Backki Kim, Jeonghwan Seo.

**Methodology:** Seung Young Lee, Zhuo Jin, Su Jang, Backki Kim, Jeonghwan Seo.

**Project administration:** Hee-Jong Koh.

**Software:** Seung Young Lee, Zhuo Jin, Su Jang.

**Supervision:** Hee-Jong Koh.

**Validation:** Seung Young Lee.

**Visualization:** Seung Young Lee.

**Writing – original draft:** Seung Young Lee.

**Writing – review & editing:** Hee-Jong Koh.

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
