## [Decision Letter · Decision Letter 0]

28 Feb 2022

PONE-D-22-02455Histological characterization of anther structure in Tetep-cytoplasmic male sterility and fine mapping of restorer-of-fertility gene in ricePLOS ONE

Dear Dr. Koh,

Thank you for submitting your manuscript to PLOS ONE. After careful consideration, we feel that it has merit but does not fully meet PLOS ONE’s publication criteria as it currently stands. Therefore, we invite you to submit a revised version of the manuscript that addresses the points raised during the review process.

We look forward to receiving your revised manuscript.

Kind regards,

MQ Shahid

Academic Editor

PLOS ONE

Journal Requirements:

3. We note that Figure 6 in your submission contain map/satellite images which may be copyrighted. All PLOS content is published under the Creative Commons Attribution License (CC BY 4.0), which means that the manuscript, images, and Supporting Information files will be freely available online, and any third party is permitted to access, download, copy, distribute, and use these materials in any way, even commercially, with proper attribution. For these reasons, we cannot publish previously copyrighted maps or satellite images created using proprietary data, such as Google software (Google Maps, Street View, and Earth). For more information, see our copyright guidelines: http://journals.plos.org/plosone/s/licenses-and-copyright.

a) You may seek permission from the original copyright holder of Figure 6 to publish the content specifically under the CC BY 4.0 license.  

Additional Editor Comments:

The results of the manuscript are very interesting, however, reviewers have raised few concerns about the manuscript that must be clarified, especially about analysis of large InDels, SVs, and if possible genetic verification by CRISPR/Cas9. So, this manuscript required a minor revision before considering for publication. Please see attached review reports for detail.

Reviewers' comments:

Reviewer's Responses to Questions

**Comments to the Author**

1. Is the manuscript technically sound, and do the data support the conclusions?

Reviewer #1: Yes

Reviewer #2: Yes

Reviewer #3: Yes

Reviewer #4: Yes

2. Has the statistical analysis been performed appropriately and rigorously? 

Reviewer #1: Yes

Reviewer #2: Yes

Reviewer #3: Yes

Reviewer #4: Yes

3. Have the authors made all data underlying the findings in their manuscript fully available?

Reviewer #1: Yes

Reviewer #2: Yes

Reviewer #3: Yes

Reviewer #4: Yes

4. Is the manuscript presented in an intelligible fashion and written in standard English?

Reviewer #1: Yes

Reviewer #2: Yes

Reviewer #3: Yes

Reviewer #4: No

5. Review Comments to the Author

Reviewer #1: The manuscript “Histological characterization of anther structure in Tetep-CMS and fine mapping of Rf gene in rice” is going to report the characterizations of Tetep-CMS in rice and their restore gene Rf. It is an interesting paper for readers. The manuscript is prepared well. The only concern of manuscript is the conclusion for candidate. There is no any experiment result to support which gene in the mapping interval is the candidate for Rf gene. Even in the discussion the authors didn’t list a number of literatures to convince us which gene must be the possible candidate for Rf. In this situation, I suggested that the author should to modify the conclusion: “suggest ORF6 annotated as pentatricopeptide repeat motif containing gene 1 ( PPR1 ), as the candidate gene responsible for fertility restoration”. In addition, the Rf gene here reported should be renamed as Rf? (Rf7 or something else, depending how many Rf genes have cloned so far in rice).

Reviewer #2: Please see attached comments for all of my revision recommendations. This is an excellent paper and very interesting research. All criteria for publication after minor revision are met. Thank you for publishing!

Reviewer #3: Comments to the Author

The group of Hee-Jong Koh employed the Tetep-CMS/ Rf lines to identify a pentatricopeptide repeat motif containing gene 1 (PPR1) in Rice. Nanopore long-read sequencing-based genome assembly to obtain genomic information and conducted fine mapping using a series of segregating populations. Genetics and phenotypic analysis of CMS plants and its downstream targets suggested that tissue-specific abnormalities in anthers are responsible for indehiscence-based sterility, and propose that the functional Rf gene is derived from allelic variation between inter-subspecies in rice. The experiments and analyses are generally conducted in a solid way, and the article is also clearly written. I believe it can meet the publication requirements after minor revision. There are three minor points I would like to address here:

1. The most important thing I am concerned with is that the Nanopore long-read sequencing of genomes were generated for identified the causal gene. This is a important because, for lots of natural mutation in plant, NGS technology such as Illumina Hiseq is not power enough to identified large InDels that lead to disrupt of the causal gene. So this should be emphasized in this paper. I think besides the discovery of an unannotated Tetep-specific segment with features of a retrotransposon within the region. The author can analyze large InDels, SVs between the three sequenced materials, and give an analysis and discussion that why other InDels and SVs is not associated with the CMS phenotype.

2. Table 1 can move to supplementary data, because it is just basic statistics for sequencing. I suggested the author show the genomic variations and their effect in the main text using table or figure.

3. The stages of anther development have clear definition and standard names in rice. Please refer to the paper from Zhang Dabing’s lab such as “Dasheng Zhang, Wanqi Liang, Changsong Yin, Jie Zong, Fangwei Gu, Dabing Zhang, OsC6, Encoding a Lipid Transfer Protein, Is Required for Postmeiotic Anther Development In Rice, Plant Physiology, Volume 154, Issue 1, September 2010, Pages 149–162, https://doi.org/10.1104/pp.110.158865”. Revised Figure 2 and related content.

One defect of this research is that they did not verify the causal gene using genetic engineering such as CRISPR/Cas9 genome-editing. If they can add this result, it would be better.

Reviewer #4: This study investigated the Histological characterization of anther structure in Tetep-cytoplasmic male sterility and fine mapping of restorer-of-fertility gene in rice. I think that the reported results are interesting. However, some more data are required.

1. In this paper, a lot of work has been done. However, please add pollen germination experiments

2. There are some typos and English style problems throughout the text that should be carefully reviewed.

6. PLOS authors have the option to publish the peer review history of their article (what does this mean?). If published, this will include your full peer review and any attached files.

Reviewer #1: No

Reviewer #2: No

Reviewer #3: **Yes: **Sha Tang

Reviewer #4: No

---

## [Author Response · Author response to Decision Letter 0]

23 Mar 2022

We appreciate the opportunity to resubmit our original research article entitled “Anatomical analysis of Tetep-cytoplasmic male sterility and fine mapping of restorer-of-fertility gene in rice” for publication in the PLOS ONE. All careful comments by reviewers are deeply appreciated, and have been addressed, with corresponding changes made directly to the manuscript where appropriate.

We ensured that our manuscript meets PLOS ONE’s style requirements. We switched our map image of Figure 6D to physical world map retrieved from The World Factbook, Central Intelligence Agency (CIA) as one of your suggested replacement source. The red circles superimposed on the map in Figure 6 is our own data which is a visual illustration of S5 Table. All figure files were processed with Preflight Analysis and Conversion Engine (PACE) digital diagnostic tool. We sincerely apologize if we misled the information regarding funding information during the submission process. Our final statement will be “This study was supported by a grant from the Next-Generation BioGreen 21 Program (No. PJ013165) of the Rural Development Administration, Korea.”

Again, we appreciate the time and effort of our editors and reviewers.

Sincerely yours

Hee-Jong Koh

---

## [Decision Letter · Decision Letter 1]

25 Apr 2022

Histological characterization of anther structure in Tetep-cytoplasmic male sterility and fine mapping of restorer-of-fertility gene in rice

PONE-D-22-02455R1

Dear Dr. Koh,

We’re pleased to inform you that your manuscript has been judged scientifically suitable for publication and will be formally accepted for publication once it meets all outstanding technical requirements.

Kind regards,

Muhammad Qasim Shahid

Academic Editor

PLOS ONE

Additional Editor Comments (optional):

Authors have done the suggested changes, so manuscript could be accepted for publication

Reviewers' comments:

Reviewer's Responses to Questions

**Comments to the Author**

1. If the authors have adequately addressed your comments raised in a previous round of review and you feel that this manuscript is now acceptable for publication, you may indicate that here to bypass the “Comments to the Author” section, enter your conflict of interest statement in the “Confidential to Editor” section, and submit your "Accept" recommendation.

Reviewer #3: All comments have been addressed

Reviewer #4: All comments have been addressed

2. Is the manuscript technically sound, and do the data support the conclusions?

Reviewer #3: Yes

Reviewer #4: Yes

3. Has the statistical analysis been performed appropriately and rigorously? 

Reviewer #3: Yes

Reviewer #4: Yes

4. Have the authors made all data underlying the findings in their manuscript fully available?

Reviewer #3: Yes

Reviewer #4: Yes

5. Is the manuscript presented in an intelligible fashion and written in standard English?

Reviewer #3: Yes

Reviewer #4: Yes

6. Review Comments to the Author

Reviewer #3: (No Response)

Reviewer #4: This study investigated the Histological characterization of anther structure in Tetep-cytoplasmic male

sterility and fine mapping of restorer-of-fertility gene in rice. In addition, The author has made revisions to the requested question, therefore, I agree to accept.

7. PLOS authors have the option to publish the peer review history of their article (what does this mean?). If published, this will include your full peer review and any attached files.

Reviewer #3: **Yes: **Sha Tang

Reviewer #4: No

---

## [Editor Report · Acceptance letter]

27 Jul 2022

PONE-D-22-02455R1 

Histological characterization of anther structure in Tetep-cytoplasmic male sterility and fine mapping of *restorer-of-fertility* gene in rice 

Dear Dr. Koh:

I'm pleased to inform you that your manuscript has been deemed suitable for publication in PLOS ONE. Congratulations! Your manuscript is now with our production department. 

Kind regards, 

on behalf of

Dr. Muhammad Qasim Shahid 

Academic Editor

PLOS ONE